# Research on China's Power Sustainable Transition Under Progressively Levelized Power Generation Cost Based on a Dynamic Integrated Generation–Transmission Planning Model

**He Huang [1,\*], DaPeng Liang [1], Liang Liang [1] and Zhen Tong [2]**

[1] School of Management, Harbin Institute of Technology, Harbin 150001, China; ldp0920@hit.edu.cn (D.L.); liangliangll88@126.com (L.L.)

[2] Electric Power Development Research Institute, China Electricity Council, Beijing 100761, China; tongzhan@cec.org.cn

[\*] Correspondence: superrio@yeah.net; Tel.: +86-10-63551015

**Abstract:** The government of China has introduced a series of energy-saving and emission reduction policies and energy industry development plans to promote the low-carbon development of the power sector. Under relatively clear and specific low-carbon development goals, the ongoing power transition has recently been studied intensively in the frame of global sustainable transition. With the development of renewable technologies, besides the long-term development goals, learning and diffusion of innovative technologies and the incentive effect of supportive policies are also important driving forces of the transition. The levelized power generation cost is the power generation cost when the net present value of the power project is zero. In this paper, the levelized power generation cost model with a learning curve and policy scenario is used to reflect the impact of technology diffusion and incentive policies from the economy perspective. By treating it as a state transfer function, a dynamic power generation–transmission integrated planning model based on the Markov Decision Process is established to describe the long-term power transition pathway under the impact of power technology diffusion and incentive policies. Through the calculation of power demand forecasting data up to 2050 and other power system information, the dynamic planning result shows that the current low-carbon policies cannot obviously reduce the expansion of coal power, but if strict low-carbon policies are implemented, the renewable power will gradually become dominant in the power structure before 2030.

**Keywords:** electric power structure; generation- and transmission-integrated planning; low-carbon police portfolio; electric power economy; China's power sector

## 1. Introduction

In the sustainable transition theory, the 'transition' of a system or industry is a process that is pushed towards long-term development goals that are more in line with collective or social interests, through continuous learning and diffusion of innovative technologies under the incentive effect of supportive policies [1]. From the perspective of global transition to a low-carbon economy, the establishment of low-carbon and green energy power systems has been the main direction of China's power development [2], and the central government and administrative authorities have designed various energy and electricity development plans, considering energy conservation, emissions reduction, and other related issues to address climate change [3]. These policies and regulations have defined the goals, phased the tasks of China's sustainable power transition, and initially established

a low-carbon development-oriented power system. With the goal of low-carbon power development, the power transition model based on the power system planning method can serve as a bridge between the power transition goal and the power system reality and is helpful to solve the problem of 'why and how to carry out' the process of power transition.

On the basis of the sustainable transition theory and power system planning method, a dynamic generation–transmission-integrated planning model considering power technologies diffusion and an incentive policy scenario is described in this paper in five sections. Firstly, the sustainable transition theory and its analysis framework are introduced, and the status quo of power system planning is discussed for the theoretical preparation of the subsequent content. Secondly, on the basis of the sustainable transition theory, the structure of the dynamic power generation–transmission-integrated planning model, in which the dynamic process is underpinned by the Markov Decision Process (MDP), is established. The objective function, state transition matrix, and constraint conditions of dynamic planning are given in the third section. In the fourth section, the long-term transition (to 2050) of China's power system under the given technology learning rate and practical information in two policy scenarios (current-level and strict low-level carbon policies) is analyzed. According to the planning results, the coal power will maintain dominance with the present technology diffusion level and the current low-carbon policy. In contrast, in the case of strict low-level carbon policy, the renewable power will gradually replace coal power and become the largest power source in the power structure after 2028. Overall, by integration of power technologies diffusion and an incentive policy scenario, the dynamic generation–transmission-integrated planning model could drive the process of power transition under the progress of power technology diffusion and policy change. In this paper, we analyze the power system sustainable transition process and the impact of incentive policy.

## 1.1. Features of Sustainable Transition

It is widely accepted that system or industrial transition is related not only to the development and progress of innovative technology itself but also to other exogenous factors regarding innovation technology development, such as market, policy, and cultural ideology [4]. Therefore, system transition cannot be analyzed only from the technological or economic perspective. In terms of research on environmental improvement and sustainable development problems, which is driven by innovative technologies evolution, the characteristics of a sustainable transition based on the 'socio-technological' paradigm theory are here summarized [5]. First, sustainable transition is usually a goal-oriented or 'purposeful' process, its target being the solution of environmental problems in sustainable development, while many other historic transitions are 'sudden' and may not have a specific transition goal (such as entrepreneurs exploring new technology-related business opportunities and so on). Second, since the goal of sustainable transition is usually to pursue the maximization of collective or social benefits, most of the solutions cannot bring obvious benefits to major stakeholders, and the tools and technologies of the solution may perform even worse than the existing current mainstream technology in terms of economy or performance. Third, the transition is often subjected to companies, enterprises, or organizations with a large complementary asset, such as automobile manufacturers, power companies, petroleum companies, logistics companies, and so on. These complementary assets not only make the existing enterprises occupy a dominant position in technology and economy but also provide the resource space for the innovation technology required by sustainable transition and accelerate the breakthrough of innovation technology. On the other hand, the inertia of large organizations will also intensify the strategic game between the established existing technology and the innovative technology.

Therefore, sustainable transition is a long-term process. Transition goal setting, transition strategy establishment, transition process control, and transition path adjustment are all problems that need to be paid attention to in the transition process [6]. Setting up and effectively promote reasonable policy measures to resolve various problems and contradictions encountered in the transition is an important means of realizing the goal of sustainable transition. The process of sustainable transition and the role of innovation technology diffusion and incentive policies are illustrated below.

As shown in Figure 1, in the initial stage of sustainable transition, the development goal may be inconsistent with the development trend of the current mainstream technology regime, and the economy of innovative technology is not obvious. In addition, the decision-making process that takes the maximization of collective interests as the decision basis will destroy the equilibrium situation of vested interests. This may easily lead to a situation in which the dominant technological and institutional infrastructure enters a free-riding or prisoner's dilemma [7]. Therefore, the degree of short-term application and diffusion of innovative technologies and positive transition incentive policies will play a key role in achieving sustainable transition goals.

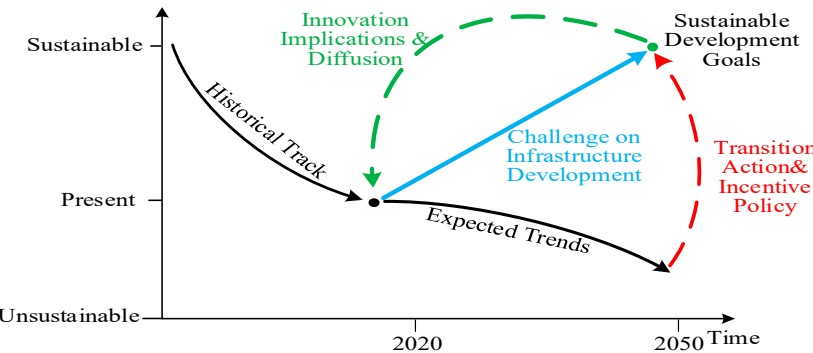

**Figure 1.** The process of sustainable transition driven by technical innovation.

### 1.2. Sustainable Transition Modelling

In the sustainable transition process of power systems driven by evolution and diffusion of renewable power generation technologies, the essence of transition is the process of evolution and diffusion of innovative renewable power generation technologies from their emergence, accumulation, and diffusion to the stage of influencing the conventional power structure and system by gradually increasing permeability [8]. During the power transition and diffusion process of renewable power innovation, there are many uncertainties, and the relationship between renewable power generation technologies, conventional power generation technologies. and corresponding incentive policies will change profoundly during the sustainable transition process. Therefore, it is highly necessary to establish a multi-stage and multi-factor nonlinear dynamic model to perform research on power system transition. The models and tools of sustainable transformation analysis are summarized in [9]; a meta-theoretical analysis framework of sustainable transition, which integrates the technology–economy and society–technology paradigms, and a policy mix analysis methods were established. The framework of the power system sustainable transition model according to the meta-theoretical analysis framework is shown in Figure 2.

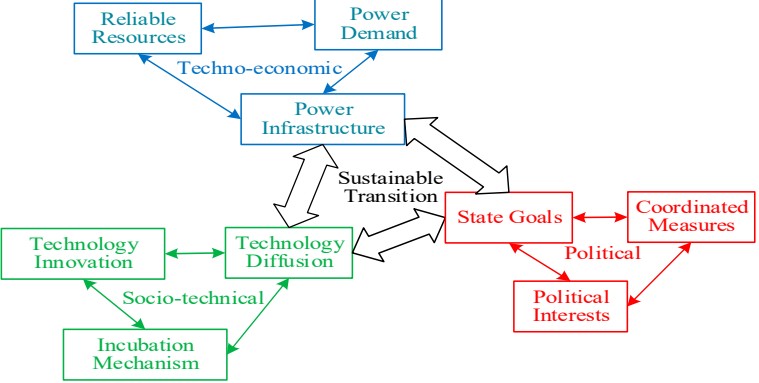

**Figure 2.** Framework of power system transition analysis.

### 1.3. Power System Transition Analysis and Modeling

Under the guidance of low-carbon development goals and incentive policies, quantitatively analyzing the sustainable transition process by a modelling/scenario study, can be a bridge between the power transition goal and the power system reality, verifying the effectiveness of low-carbon transition policies and measures and promoting the transition process. There have been some researches focused on power system transition modelling/scenarios analysis. However, most models used a top-down approach, analyzing the power transition pathway from the perspective of energy economy [10–12] or emission reduction index decomposition [13,14] at the macro level. These studies take the power system as a link of the energy system, simplifying the process of power system development and power technology evolution, which may be out of the actual condition of the power system and also not reflect the specific impact of technological progress and incentive policies. By the power system planning method, which establishes the bottom-up power transition model, it could be easier to describe and analyze the process of power system transition by adjusting the parameters or setting different scenario.

Traditional power system planning is conducted on the basis of identified load forecasting and searches for an optimal decision scheme of power system extension [15]. Therefore, the identified load information can be seen as the border of the planning approach, and the power generation extension planning (GEP) and transmission extension planning (TEP) as the core. GEP and TEP have been widely used separately to solve meso and micro specific projects of power operations, but this situation has largely led to the curtailment of renewable energy and a low installed generation utilization rate in the power sector [16]. Therefore, by combining the GEP and TEP models together and enlarging the planning objectives to minimize social investment or maximize environmental benefits, it is possible to establish the integrated generation–transmission extension planning (IG-TEP) model that can be used to solve problems in the power system macro development and transition.

The development and current status of GEP and TEP are reviewed in [17,18], either alone or in terms of their integration. After a detailed comparison, the authors emphasized the importance of IG-TEP. Reference [19] put forward a coordinated planning model based on interactive and iterative processes between the GEP and the TEP. The obvious economic advantages of an integrated model over a separated planning were proven in [20–22] by analyzing the IG-TEP model. The authors presented a multistage stochastic programming model of IG-TEP in [23] to address sustainable problems and the uncertainties of future electricity demand, fuel prices, and greenhouse gas emissions to which the power system was subjected. A linear optimization model was built in [24] to define cost-optimal pathways toward a sustainable power system in the Association of East Asian Nations (ASEAN), and the results suggested to foster a diversified extension of renewables and elaborating on political and technical solutions. An IG-TEP model was introduced in [25], where the authors argued that it was necessary to identify a long-term macro-level integrated expansion plan for the total power sector. A probabilistic model for the IG-TEP problem considering a reliability criterion was described in [26]. A three-level equilibrium IG-TEP model was built in [27]. Concerns about low-carbon emissions, security, and economical energy supply are pushing various countries to consider a strategic energy power planning [28].

Many studies have proven the feasibility of IG-TEP, and several studies have even set the methodology and applications for macro-level analyses. However, in most previous IG-TEP studies based on static planning with all parameters fixed, it was difficult to truly and accurately describe or evaluate the process of power transition, since power transition is a purposeful and long-term process affected by the changes and uncertainties derived from power technology evolution and incentive policies adjustments. Therefore, on the basis of the sustainable transition theory, this paper establishes a dynamic IG-TEP model based on the MDP to describe the transition path of China's power structure. The long-term changes of China's power structure, power grid layout, economy of construction and operation, and carbon and pollutant emission under the influence of power technology evolution and incentive policy are analyzed accordingly.

## 2. Dynamic IG-TEP Based on Sustainable Transition Theory

According to the sustainable transition theory, power system transition is affected by the speed of innovation technologies diffusion, which may bring about endogenous changes of a technical economy in every step from generation to distribution. In addition, the development goals set by the government and innovation incentive and support policies will affect the tendency of participants in the planning scheme and the direction of investment decisions. Therefore, under the influence of innovation technology diffusion and incentive policies, the processes of power system transition under a specific goal is equivalent to a dynamic programming problem with a known initial and final state and an uncertain intermediate process. Therefore, from the perspective of sustainable transition, considering the dynamic effects of technology evolution/diffusion mechanisms and low-carbon development incentive policies, the IG-TEP method could be used to analyze and model the dynamic transition process. The structure of the dynamic IG-TEP model considering power technology learning and incentive policies is shown in Figure 3.

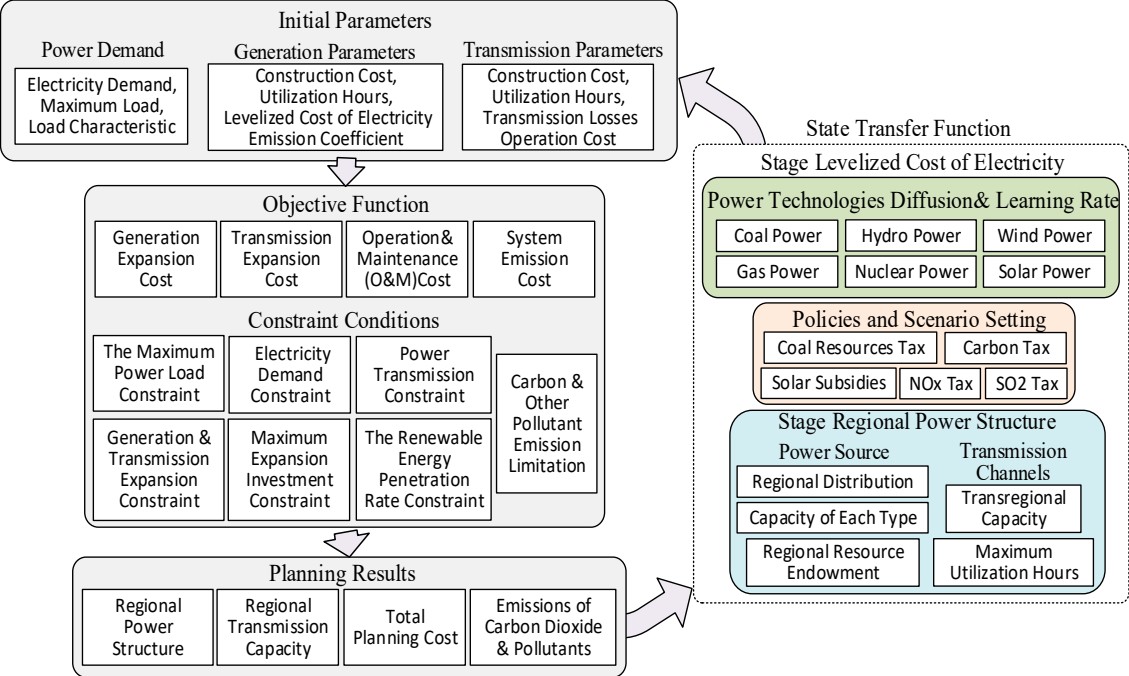

**Figure 3.** The main idea of the dynamic power-grid integrated planning model.

As shown in Figure 3, the dynamic process of the model is represented by the updated state transition matrix at each stage, which is produced by a levelized cost of power generation, considering power technology learning and incentive policy. By the state transfer function matrix of the MDP and the planning mechanism of the IG-TEP model, the changes of long-term power structure will be optimized according to a given power demand and related information Hence, the dynamic IG-TEP model based on the sustainable transition theory is composed of the planning mechanism and a dynamic state transition function. Here, the dynamic state transition function reflects the technologies diffusion and policies changes. The power innovation diffusion and policy changes confer the processes uncertainty and irreversibility. The uncertainty of planning is because the time value of capital, technology diffusion, policy mix, and renewable power permeability vary from stage to stage, which means that even the same planning scheme combination will produce different values at different stages. Thus, the necessity of dynamic planning emerges due to the subsequent decisions, and the target function value will change accordingly. As long-time power planning must be sequential, the state of a stage is determined by the previous stage, so the entire power planning process can be regarded as irreversible.

## 3. Dynamic Power Generation and Transmission Integrated Planning Model

### 3.1. Dynamic Planning Processes Based on Markov Decision Process (MDP)

The structure and planning mechanism of the dynamic IG-TEP model are described above. In order to establish a complete dynamic planning processes, the state transition matrix needs to be defined to connect the different stages of the planning. From the aspect of the state transition function, the multi-stage decision process in the planning model was treated as an MDP [29]. In this paper, the power planning period was divided into several stages, and each stage was optimized according to the stage planning information and parameters. In the process of optimization, the state transition is reflected by the state transfer probability matrix, which is formed from the input data and relevant parameters under the influence of the policy portfolio at a single stage. One matrix corresponds to one optimized generation and transmission scheme, which describes the influence of different policy portfolios and different construction and operation costs of power technologies from the current state to the next state. Therefore, the pathway of the optimal scheme within the whole period can be established by iterating the new state set in the next planning stage until the end of the planning period. A schematic of optimal processes in dynamic power planning is shown in Figure 4.

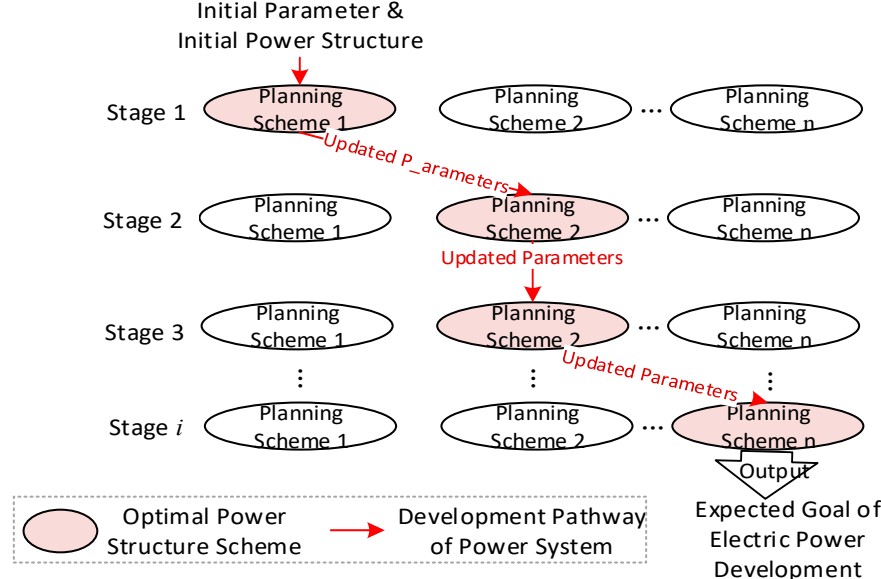

**Figure 4.** State transfer process of the dynamic planning model.

### 3.2. Objective Function

MDP assumes that each action can be completed in a decision period, then the dynamic programming is solved by the optimization options of multiple state transitions. Therefore, power planning based on MDP can be explained as the power system periodically extracting environmental information: first interacting with the environment for more complete state observations and decisions selection, then performing an action on a specific state in the current environment, and finally moving to the next environment state to begin a new decision cycle. In general, such an MDP can be represented by a multivariate group, that is:

$$\left( S,\ A_x,\ f(x, \pi(x)),\ P^{\pi(x,i)}(x, x'),\ \pi(x, i) \right),$$

where $S$ represents a finite set of states (current state of the system); $A_x$ is the decision set corresponding to the state $x \in S$; $f(x, \pi(x))$ is the return function for strategy $\pi \in A_x$ on the state $x \in S$; $P^{\pi(x,i)}(x, x')$ is the transfer matrix for state $x \rightarrow x'$ by strategy $\pi \in A_x$ on the state $x \in S$; and $\pi(x, i)$ is the strategy from $A_x$ in state x at moment $i$ [30].

Therefore, for a dynamic generation–transmission-integrated planning model, in this paper, the stage in the planning process was denoted by $i$, where $i = 1, 2, \ldots, I$. So, firstly, the planning stages in this paper covered the period from 2018 to 2050; we used 2018–2020 as the first stage, then considered a new planning stage every five years, assuming that the annual power investment was completed at the beginning of the year. Secondly, the state set of the power system at stage $i$ is $S$ and a possible state is denoted as $x \in S$; the solution to the stage planning is represented by strategy $\pi$ ($\pi : S \to A$), which is the basic element that makes up a decision set $A_x$ and also reflects a mapping from the states set to the actions set. If the policy set $A_x$ at stage $i$ is adopted, $\pi_{ix} \in A_x$ corresponds to a strategy from the decision set $A_x$. Then, the value function of a Markov programming model for the power generation and grid integrated planning can be expressed as:

$$V^\pi(x) = \min\left[\sum_{i=0}^{I-1} f(x_i, \pi(x_i))\Big| x_0 = x\right]$$

$E$ is used to describe the total social cost expectations of state $x$ in the strategy $\pi$, during the sub-state evolution from $s \to f$. For $i = 0$, we indicated the initial state $x_0$. Thus, the evolution process of the system from state $x$ to state $x'$ could be described by a state transition function $T(x, \pi, x')$, Therefore, the process of the system state change could be defined by the previous value function, and the recursive equation for $V^\pi(x)$ is as follows:

$$V^\pi(x) = f(x, \pi(x)) + \sum_{x \in S} T(x, \pi, x') V^\pi(x')$$

For a power generation and grid integrated planning model:

$$f(x, \pi(x)) = \min \left\{ \sum_{t=1}^{T} \left[ \sum_{r=\Omega_r} \sum_{s=\Omega_s} (Cs_{s,r,t} - (Cs_{s,r,t-1} - Q_{s,r,t})) P_S \right.\right.$$
$$+ \sum_{r=\Omega_r} \sum_{s=\Omega_s} Cs_{s,r,t} H_s LCoE_{x,n} + \sum_{r=\Omega_r} \sum_{s=\Omega_s} \varepsilon c_{s,t} Cs_{s,r,t} H_s Pr_C$$
$$+ \sum_{r=\Omega_r} \sum_{s=\Omega_s} \varepsilon s_{s,t} Cs_{s,r,t} H_s Pr_S + \sum_{r=\Omega_r} \sum_{s=\Omega_s} \varepsilon n_{s,t} Cs_{s,r,t} H_s Pr_N + \sum_{g \in \Omega_g} \left( Cg_{g,t} \right.$$
$$\left.\left.\left. - Cg_{g,t-1} \right) P_g \right] \frac{1}{(1+R)^t} \right\},$$

where $T$ is the planning period; $\Omega_r$ represents the set of regions contained in the planning; $\Omega_s$ represents the type of power source; $Cs_{s,r,t}$ is the capacity of power source $s$ in region $r$ at planning stage $t$; $Q_{s,r,t}$ is the retired capacity of power source $s$ in region $r$ at planning stage $t$; $P_S$ is the installation cost per unit capacity of source $s$; $H_s$ is the annual utilization in hours of source $s$; $LCoE_{x,n}$ is the levelized cost of electricity; $\varepsilon c_{s,t}$, $\varepsilon s_{s,t}$, and $\varepsilon n_{s,t}$ are the carbon dioxide, sulfur dioxide, and nitrogen oxide emission coefficients of power source $s$ at stage $t$, respectively; $Pr_C$ is the unit carbon tax price; and $Pr_S$, $Pr_N$ are the effluent charges of sulfur dioxide and nitrogen oxide, respectively. $\Omega_g$ is the set of transmission channels; $Cg_{g,t}$ is the transmission capacity of channel $g$ at stage $t$; $P_g$ is the construction cost of $g$; and $R$ is the discount rate [31]. This equation gives the staged objective function of the dynamic model. There are six elements in brackets: the first one is the planning costs of the power sources expansion, the second one represents the operation and maintenance costs of power source production under staged power structure, the next three ones represent the carbon and pollutant emission costs, and the last one is the costs of power grid expansion.

The above function describes the planning mechanism of the power-grid integrated planning model, and the dynamic processes are reflected by the state transition matrix, which expresses the change of the time value of capital, technology diffusion, policy mix, and renewable power permeability during each planning stage. Thus, $LCoE_{x,n}$, here, is equivalent to the state transition function $T(x, a, x')$

in MDP, which indicates the varying feed-in price of generation technologies that is caused by the action of technology diffusion and policy scenario in different stages.

*3.3. State Transfer Function Considering Technology Learning and Incentive Policy*

In this paper, the state transition matrix is reflected by the variation of the $(LCoE_{x,n})$ of each power generation technologies, which is influenced by the technology diffusion and policy scenarios in each stage. The $LCoE_n$ of an electric power project is the power generation cost, which achieves the lowest expected rate of return, that is, the net present value of the power project is zero. Therefore, the $LCoE_s$ of power technology can be expressed as:

$$LCoE_n = \left( \sum_{n=0}^{N} \frac{Cost_n}{(1+r)^n} \right) \Big/ \left( \sum_{n=0}^{N} \frac{E_n}{(1+r)^n} \right)$$

where $Cost_n$ is the construction cost of the power project, $N$ is the operation period of the project, $r$ is the discount rate. Thus, it can be seen from the above equation that the present value of the $LCoE_n$ times the annual electric power output $E_n$ equals the present value of the project expenditure.

Therefore, it is possible to divide the cost of power generation project into three parts, i.e., construction cost, operation cost, and external cost. The construction cost includes the land cost, equipment cost, installation cost, relative taxes during the construction period, etc. Operation and maintenance (O&M) costs include production costs, maintenance costs, fuel costs, and various taxes and fees during the operation stage. The external cost of in this paper mainly refers to the economic loss of the external environment caused by output contaminants such as carbon dioxide, sulfur dioxide, nitrogen oxides, and other pollutants in the power generation process. On this basis, $LCoE_x$ considering the impact of externalities of power production and low-carbon policy is as follows:

$$LCoE_{x,n} = \left( \sum_{n=1}^{N} \frac{(CC_{n,x}CT_{n,x} + OM_{n,x}OT_{n,x} + EC_{n,x})}{(1+r)^n} \right) \Big/ \left( \sum_{n=1}^{N} \frac{\left( Cap_x H_{n,x} \left( 1 - o_\eta \right) \right)_n}{(1+r)^n} \right)$$

where $n$ is the year of planning, $n \in N$, and $N$ is the life span of the power project; $x$ is the type of power generation technology; $CC_{n,x}$ is the annual value of unit investment cost during the construction period of power technology $x$, under the unit installation cost price in year $n$th; $CT_{n,x}$ is the taxes in the construction period of power technology $x$, under the unit installation cost price in year $n$th; $OM_{n,x}$ is the operation and maintenance cost of the power technology x in production at the $n$th year price level, and $OT_{n,x}$ is taxes related to the operation and maintenance in production of the power technology $x$ in the policy scenario of year $n$th; $EC_{n,x}$ is the external cost of the power technology $x$ in production in the policy scenario of year $n$th; $Cap_x$ is the newly installed capacity of the power technology $x$ in year $n$th, and $H_{n,x}$ is the average utilization hours of the power technology $x$ in year $n$th.

In the calculation of the construction costs, the annual value of the unit capacity and the installation investment cost of power technology account for the largest proportion. Thus, in the calculation of the electricity generation cost, considering the time value of capital, the equivalent annuity approach was used to divide the initial investment into each year. Hence, the annual value of the construction cost of power technology including the taxes couls be calculated as follows:

$$CC_{n,x}CT_{n,x} = (C_{n,x}Cap_x + C_{n,x}Cap_xCT_{n,x})APV_x$$

$$APV_x = \frac{r(1+r)^N}{(1+r)^N - 1}$$

where $C_{n,x}$ is the unit investment cost of power technology $x$, under the unit installation cost price in year $n$th; $APV_x$ is the present value coefficient of annuity, and $r$ is the discount rate.

The calculation equation of annual O&M cost of power technology $x$ is:

$$OM_{n,x}OT_{n,x} = Cap_xH_{n,x}b_{C,x}Pc_{,x} + \left(\frac{Cap_xH_{n,x}}{1 + OMF_{n,x}}\right)(1 + OT_x)$$

where $b_{C,x}$ is the fuel coefficient of the power technology $x$, $Pc_{,x}$ is the price of the fuel, $OMF_{n,x}$ is the other fixed operation cost coefficient, $OT_x$ is the taxes relevant to electricity generation and O&M processes.

For the external cost, the cost of environmental pollution comes in the form of taxes on pollutants in the electricity production process, such as carbon tax and pollutant emission tax. It is calculated as follows:

$$EC_{n,x} = (H_{n,x}Cap_x)b_{C,x}RT_C + \eta_{m,x}(H_{n,x}Cap_x)ET_m$$

where $RT_C$ is the tax on fossil energy resources determined by the policy scenario, $ET_m$ is the pollutants emission taxes, and $\eta_{m,x}$ is the pollutant discharge coefficient of the power technology $x$.

Finally, the learning curve usually describes the trend of power technology diffusion. Therefore, in order to describe the development process of the power system more accurately, the signal factor learning curve was incorporated into the $LCoE_s$ equation.

If the diffusion trend of power technology $x$ in stage n can be represented by the learning curve $SC_{n,x}$, the formula for calculating the annual value of the initial investment cost of power technology $x$ with the learning curve is:

$$CC_{n,x}CT_{n,x} = [SC_{n,x}Cap_x + SC_{n,x}Cap_xCT_{n,x}]APV_x$$

$SC_{n,x}$ can be decomposed into:

$$SC_{n,x} = SC_{0,x} + (SC_{1,x} - SC_{0,x})(Cap_x - Cap_{0,x})^{-a}$$

where $SC_{1,x}$ is the initial cost of the power technology $x$ in stage $n$, $SC_{0,x}$ is the minimal cost of the power technology $x$ in stage $n$, $Cap_{0,x}$ is the initial capacity of the power technology $x$ in planning stage, and $a$ is the learning rate in 'learning by doing'.

The diffusion of power technology will have a great impact on the construction investment and power generation cost. By setting up the state transfer function, which reflects the power technology diffusion and incentive policies scenario, the dynamic power system integrated planning model could be established.

*3.4. Constraint Conditions*

After determining the objective function, the dynamic power planning model based on MDP also included several constraints, such as power demand and load reliability constraints, social resource reserves, economic and technological development constraints, annual and total carbon or other pollutant emission limitation constraints, and so on.

a.     The maximum power load constraint ensures that the peak load demand in each region for each year can be met. This means that the power balance at the non-peak hours can also be achieved during the entire planning period. This constraint is formally expressed as:

$$\left(\sum_{s\in\Omega_s} Cs_{s,r,t} - \sum_{g\in\Omega_{r\to x}} Pt_{g,t} + \sum_{g\in\Omega_{x\to r}} Pt_{g,t}\left(1 - l_g\right)\right)(1 - l_r) \ge Fp_{r,t}(1 + \mu)(r \in \Omega_r, \ t = 1,2,3,\ldots,T),$$

where $\Omega_{r\to x}$ is the set of transmission channels in region $r$ that export electric power to other regions; $\Omega_{x\to r}$ is the set of transmission channels in region $r$ that import electric power from the outside regions; $Pt_{g,t}$ is the power transmitted by channel $g$ at stage $t$; $l_g$ is the line loss rate of

channel $g$; $l_r$ is the average line loss rate of the grid within region $r$; $Fp_{r,t}$ is the forecasted peak load in region $r$ at stage $t$; and $\mu$ is the reserve factor [32].

b.　The electricity demand constraint is to guarantee that the yearly accumulative electricity balance can be realized:

$$\left( \sum_{s=\Omega_s} Cs_{s,r,t}H_s - \sum_{g\in\Omega_{r\to x}} Et_{g,t} + \sum_{g\in\Omega_{x\to r}} Et_{g,t}\left(1-l_g\right) \right)(1-l_r) \geq Fp_{r,t}$$
$$(r \in \Omega_r,\ t = 1,2,3,\dots,T)$$

where $Et_{g,t}$ is the annual electricity transmitted by channel $g$, and $Fp_{r,t}$ is the forecasted electricity demand in region $r$ at stage $t$.

c.　The power transmission constraint is $Pt_{g,t} \leq Cg_{g,t}$.

d.　The annual electricity transmission constraint is $Et_{g,t} \leq Cg_{g,t}H_g$, where $H_g$ is the annual maximum utilization in hours of channel $g$.

e.　The generation and transmission expansion capacity constraints are:

$$0 \leq Cs_{s,r,t} - Cs_{s,r,t-1} + Q_{s,r,t} \leq Csm_{s,r,t} \quad 0 \leq Cg_{g,t} - Cg_{g,t-1} \leq Cgm_{g,t},$$

respectively. Where $Csm_{s,r,t}$ represents the maximum expansion capacity of power source $s$ in region $r$ at stage $t$, and $Cgm_{g,t}$ is the maximum expansion capacity of channel $g$ at stage $t$.

f.　The generation expansion investment constraint is:

$$\sum_{r\in\Omega_r}\sum_{s\in\Omega_s} (Cs_{s,r,t} - (Cs_{s,r,t-1} - Qs_{s,r,t}))P_s \leq I_s,$$

where $I_s$ is the upper limit of the investment amount on generation expansion. The transmission expansion investment constraint is:

$$\sum_{g\in\Omega_g} \left(Cg_{g,t} - Cg_{g,t-1}\right)P_g \leq I_g,$$

where $I_g$ is the upper limit of the investment amount on transmission expansion.

g.　The renewable energy penetration constraint is:

$$\frac{\sum_{r\in\Omega_r}\sum_{s\in\Omega_{sr}} Cs_{s,r,t}}{\sum_{r\in\Omega_r}\sum_{s\in\Omega_s} Cs_{s,r,t}} \geq \theta_t,$$

where $\Omega_{sr}$ is the set of the renewable energy power source, and $\theta_t$ is the maximum penetration rate at each stage.

h.　The annual carbon emission and other pollutant emission limitation constraints are:

$$\sum_{r\in\Omega_r}\sum_{s\in\Omega_s} \varepsilon c_{s,t}Cs_{s,r,T}H_s \leq Uce_T$$

$$\sum_{r\in\Omega_r}\sum_{s\in\Omega_s} \varepsilon s_{s,t}Cs_{s,r,T}H_s \leq Use_T$$

$$\sum_{r\in\Omega_r}\sum_{s\in\Omega_s} \varepsilon n_{s,t}Cs_{s,r,t}H_s \leq Une_t,$$

where $Uce_t$, $Use_T$, $Une_t$ are the annual limits for carbon dioxide, sulfur dioxide, and nitrogen oxide emissions, respectively.

i.      Finally, the total cumulative pollutant emission constraints are:

$$\sum_{t=1}^{T} \sum_{r\in\Omega_r} \sum_{s\in\Omega_s} \varepsilon c_{s,t} Cs_{s,r,t} H_s \leq Uce$$

$$\sum_{t=1}^{T} \sum_{r\in\Omega_r} \sum_{s\in\Omega_s} \varepsilon s_{s,t} Cs_{s,r,t} H_s \leq Use$$

$$\sum_{t=1}^{T} \sum_{r\in\Omega_r} \sum_{s\in\Omega_s} \varepsilon n_{s,t} Cs_{s,r,t} H_s \leq Une,$$

where, *Uce*, *Use*, *Une* are the maximum cumulative emission amounts for carbon dioxide, sulfur dioxide, and nitrogen oxide over the entire planning period.

*3.5. Calculation Method*

There are many algorithms to solve dynamic programming problems. Particle swarm optimization (PSO) is widely used to solve dynamic programming problems because of its advantage of fast convergence speed. However, due to the PSO is easy to trap into the local solution, in this paper, a cultural algorithm was embedded in the PSO to remedy for this shortcoming. Because the length of the article is limited, the details of the cultural-based particle swarm optimization (CBPSO) algorithm will not be discussed here. This section will only introduce the main procedure and basic logic of power generation and transmission expansion planning through CBPSO.

In the calculation process of CBPSO, the generation and transmission expansion schemes are regarded as the particles of PSO; thus, in the optimization process, each particle is tracked for its individual historical optimal solution and global historical optimal solution, but only the best particles are put into a belief space of cultural algorithm. Through the evaluation of situation knowledge and standardized knowledge based on power planning constraints and power development target in the belief space, the optimal particles that meet the conditions continue to the optimal process, and the particles that do not meet the conditions will be adjusted and optimized by situation knowledge and standardized knowledge. This operation will continue to be looped until the end of the entire planning period. According to this procedure, the optimization processes of the CBPSO are shown Figure 5 below.

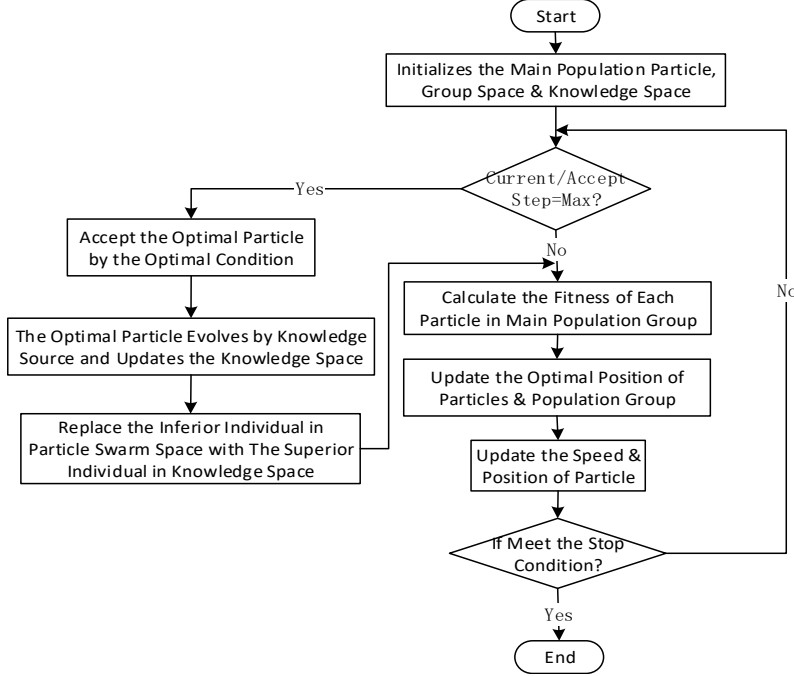

**Figure 5.** Optimization processes of the cultural-based particle swarm optimization (CBPSO).

## 4. Influences of a Low-Carbon Policy on the Development of China's Power Sector

### 4.1. Base Year Electric Power Structure by Grid Regions

To analyze the development of China's power sector from 2018 to 2050, it is first necessary to describe the current state of China's power system. Figure 3 shows the regional division of China's power grid and the currently installed composition of the power sources in these regions. According to the jurisdiction covered by the power grid, China can be divided into six grid regions: northeast, north, central, east, northwest, and south [33,34]. For the installed structures, it can be seen that coal-fired power plants remain the main power source in all regions, accounting for over 50% of the total generated power. Following coal, the installed structures from large to small are hydro power, wind power, solar power, and nuclear power. In addition, China's energy pattern is characterized by the distribution of energy resources in northwest, north, and central China, and the concentration of power demand in central, eastern, and southern China. To deal with this problem, eight transmission channels were constructed to connect the six grid regions. Furthermore, the southern and eastern regions have a large load demand, and western and northern China are rich in energy resources, so the transmission pattern in China presents a tendency of power transmission from west to east [35,36]. The information about the installed power sources and regional connections (shown as arrows in the figure) for the model in this paper is based on the structure in Figure 6.

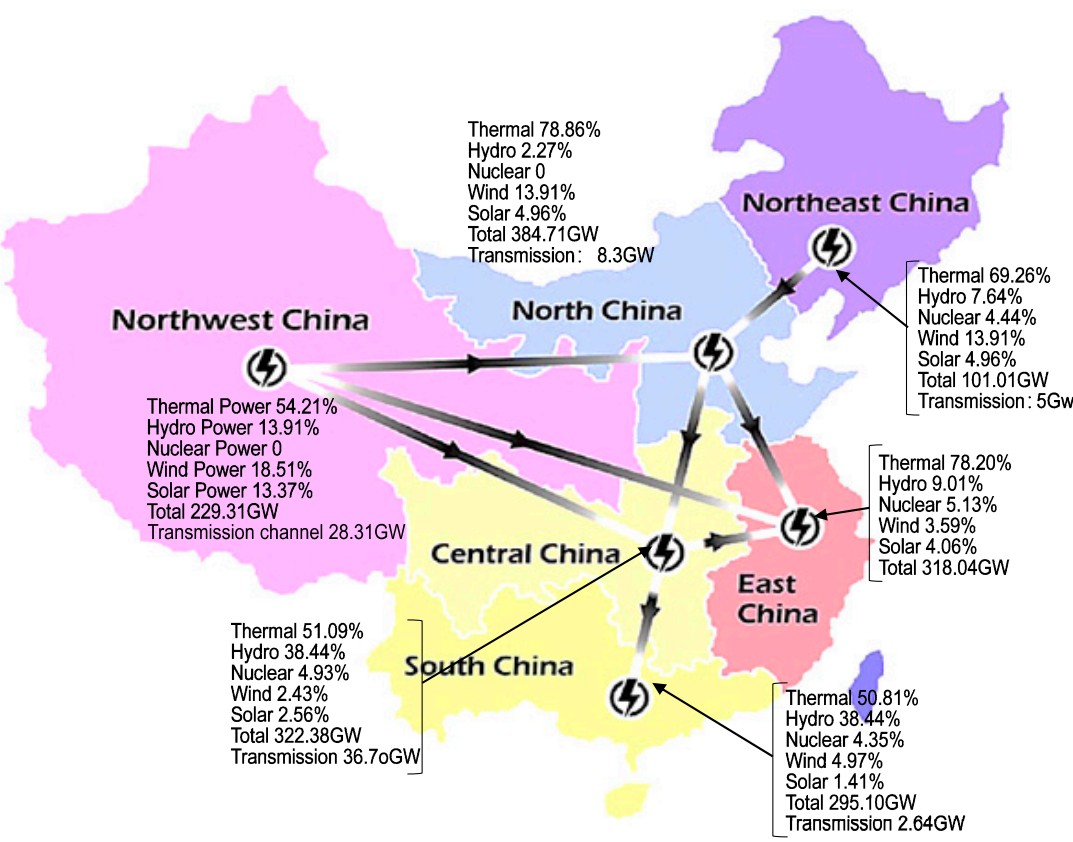

**Figure 6.** Regional division of China's power grid and corresponding installed structures.

### 4.2. Input Data and Parameters

The IG-TEP in this paper included six power resources: coal-fired, gas, hydro, nuclear, wind, and solar. Their initial planning parameters are shown in Table 1.

**Table 1.** Initial planning parameters of the power source and transmission lines. O&M: operation and maintenance.

| Name of Equipment | Construction Cost (RMB/kW) | Life Span (Years) | Utilization (h) | Fixed O&M Cost (RMB/kWh) | Exploitable Resources (GW) | Annual Maximum Exploitable (GW) |
|---|---|---|---|---|---|---|
| Coal Power | 3300 | 30 | 5500 | 0.66 | 14,000 | 400 |
| Hydro Power | 3800 | 40 | 4000 | 0.7 | 10,000 | 500 |
| Gas Power | 4200 | 25 | 7000 | 1.8 | 6000 | 200 |
| Nuclear | 6000 | 50 | 3500 | 0.75 | 5416 | 200 |
| Wind Power | 6800 | 20 | 2000 | 0.4 | $5 \times 10^5$ | 2000 |
| Solar Power | 7500 | 20 | 1800 | 0.4 | $1.86 \times 10^8$ | 4000 |
| Transformer | 5500 | 30 | 7500 | 3.5 | - | - |

For fired generators (coal-fired power source and gas power), the emissions characteristics and total amount of pollutants should be limited. The emission coefficient and annual emission limitations of fired-power sources are shown in Table 2. The data and parameters in Table 2 were obtained from References [37,38], and the units of parameters were unified.

**Table 2.** Emission coefficients and annual emissions limitations for fired-power sources.

| Power Source Type | Coefficient Category | 2017 | 2020 | 2030 | 2040 | 2050 |
|---|---|---|---|---|---|---|
| Coal Power Emission Coefficient | $CO_2$ Emissions (g/kWh) | 800 | 800 | 700 | 650 | 600 |
| | $SO_2$ Emissions (g/kWh) | 4.3 | 4.3 | 4.0 | 3.8 | 3.6 |
| | $NO_x$ Emissions (g/kWh) | 2.7 | 2.7 | 2.6 | 2.5 | 2.3 |
| Gas Power Emission Coefficient | $CO_2$ Emissions (g/kWh) | 430 | 430 | 420 | 410 | 400 |
| | $SO_2$ Emissions (g/kWh) | 0 | 0 | 0 | 0 | 0 |
| | $NO_x$ Emissions (g/kWh) | 1.9 | 1.9 | 1.7 | 1.5 | 1.3 |
| Total $CO_2$ Emissions Cap ($\times 10^9$ tons) | | 48 | 50 | 55 | 55 | 55 |
| Total $SO_2$ Emissions Cap ($\times 10^9$ tons) | | 0.12 | 0.13 | 0.14 | 0.16 | 0.16 |
| Total $NO_x$ Emissions Cap ($\times 10^9$ tons) | | 0.40 | 0.40 | 0.49 | 0.54 | 0.56 |

The learning rate of different power technologies in the medium and long term and corresponding unit investment costs are predicted in [39], on which basis, the learning rate and unit investment cost of different power technologies from 2017 to 2050 could be calculated and are shown in the following Table 3.

**Table 3.** Average learning rate and investment cost of power sources in different period.

| Power Source Type | 2020–2030 | | 2030–2040 | | 2040–2050 | |
|---|---|---|---|---|---|---|
| | ALR | ICpU | ALR | ICpU | ALR | ICpU |
| Coal Power | 1.20 | 3560 | 0.30 | 3450 | 0.10 | 3400 |
| Hydro Power | 1.30 | 3380 | 0.60 | 3250 | 0.30 | 3200 |
| Gas Power | 0.30 | 11,200 | 0.00 | 10,750 | 0.00 | 10,500 |
| Nuclear | 5.70 | 14,500 | 3.10 | 12,800 | 1.60 | 12,000 |
| Wind Power | 6.44 | 7600 | 3.40 | 6800 | 2.30 | 6250 |
| Solar Power | 10.10 | 9650 | 5.62 | 8200 | 4.10 | 7400 |

ALR: Average Learning Rate (%); ICpU: Investment Cost per Unit (Yuan/kW).

The parameters used to calculate the $LCoE_x$ included financial variables, labor costs and various taxes and other expenses. The set values are shown in the Table 4.

**Table 4.** Financial parameters for LCoEx calculation.

| Financial Parameter | Values | Fixed Taxes Rate | Values |
|---|---|---|---|
| Equity Fund Rate (%) | 20 | Income Tax (%) | 25 |
| Length of Maturity/a | 15 | Added-value Tax (%) | 17 |
| Annual Interest Rate (%) | 5 | Land Use Tax (%) | 1.2 |
| Residual Asset Value (%) | 5 | Urban M&O Tax (%) | 5 |
| Depreciation Rate (%) | 5 | Additional Tax of Education (%) | 1 |
| Depreciable Life /a | 20 | Fuel Input Tax (%) | 13 |
| Internal Rate of Return on Capital (%) | 8 | Material Input Tax (%) | 17 |
| Discount Rate (%) | 8 | Other Fixed Cost Taxes (%) | 18 |

After determining the planning parameters and operation coefficients needed to calculate the system normally, it was necessary to express the change of power demand in the planning period. The authors of Reference [40] described the change in China's future power demand. With the large-scale development of clean power, the progress of electricity technology, and the improvement in the power market, the competitiveness of electric power in energy usage terminals has been significantly enhanced and is driving the continuous growth of power demand. Therefore, in the planning period from 2018 to 2050, China's total electricity demand will continue to grow, but the growth rate will gradually slow down, remaining above 3% until 2030 and reaching a saturation stage around 2035. Then, it will maintain 1–2% growth from 2035 to 2045 and drop to about 1% by 2050. The total electric power demand will reach 12–14 TWh. On basis of these predictions, the forecasting of China's power demand from 2018 to 2050 is shown in Table 5.

**Table 5.** Forecasting of China's regional power demand from 2018 to 2050.

| (TWh) | 2018 | 2020 | 2025 | 2030 | 2035 | 2040 | 2045 | 2050 |
|---|---|---|---|---|---|---|---|---|
| Northeast | 400 | 434 | 492 | 552 | 580 | 610 | 617 | 624 |
| North | 1249 | 1369 | 1663 | 1918 | 2061 | 2210 | 2302 | 2391 |
| Central | 915 | 1034 | 1285 | 1527 | 1754 | 2117 | 2399 | 2505 |
| East | 1989 | 2289 | 2611 | 2794 | 2928 | 3028 | 3064 | 3150 |
| Northwest | 850 | 937 | 1125 | 1288 | 1512 | 1629 | 1740 | 1814 |
| South | 1394 | 1434 | 1622 | 1819 | 1963 | 2003 | 2075 | 2115 |
| Total Electricity Demand | 6800 | 7500 | 8800 | 9900 | 10,800 | 11,600 | 12,200 | 12,600 |

*4.3. Scenario Settings*

In the context of China's power development from 2018 to 2050, the following two scenarios were set according to the intensity of the energy saving and emission reduction policies:

Scenario 1: Planning starts from 2018, and the planning period is from 2018 to 2050. The power sources considered include coal-fired, gas, hydro, nuclear, wind, and solar sources. The basic energy savings and emissions reduction policy are maintained, and a carbon tax and renewable energy price subsidy is not implemented.

Scenario 2: Planning starts from 2018, and the planning period is from 2018 to 2050. The power sources considered include coal-fired, gas, hydro, nuclear, wind, and solar sources. A stringent energy-saving and emissions reduction policy is implemented; a coal resource tax, carbon tax, and other pollutant emissions taxes are set at a high level, and an electricity price subsidy for renewable energy generation is encouraged.

In the analysis below, S1 and S2 were used to replace Scenarios 1 and 2, respectively, and the intensity of energy-saving and emissions reduction policies and pollutant emissions tax settings in S1 and S2 are shown in Table 6.

**Table 6.** Energy-saving and emissions reduction policies settings in Scenarios S1 and S2.

| | Normal Policy Portfolio (S1) | | | | | Strict Policy Portfolio (S2) | | | | |
|---|---|---|---|---|---|---|---|---|---|---|
| | 2018 | 2020 | 2030 | 2040 | 2050 | 2018 | 2020 | 2030 | 2040 | 2050 |
| Coal Resources Tax (%) | 4 | 4 | 6 | 6 | 6 | 10 | 12 | 14 | 16 | 18 |
| SO$_2$ Emissions Tax (RMB/Tons) | 1 | 1 | 1 | 1 | 1 | 3 | 4 | 5 | 6 | 7 |
| NO$_x$ Emissions Tax (RMB/Tons) | 0.8 | 0.8 | 0.8 | 0.8 | 0.8 | 2.8 | 3.8 | 4.8 | 5.8 | 6.8 |
| Carbon Tax (RMB/Tons) | 0 | 0 | 0 | 0 | 0 | 0 | 50 | 80 | 100 | 120 |
| Solar Subsidies (RMB/kWh) | 0 | 0 | 0 | 0 | 0 | 0.4 | 0.35 | 0.3 | 0.25 | 0.22 |

*4.4. Results and Analysis*

The changes in China's power structure from 2018 to 2050, as calculated by our model, are shown in Figures 5 and 6.

Figure 7 shows the results of S1, which is the scenario where the basic energy savings and emissions reduction policy remain the same as in 2017. Furthermore, S1 implementes no carbon tax or renewable energy price subsidy. The results in Figure 4 show that the S1 policy set has little effect on the large-scale development of coal power, as the cost of generating electricity from coal is still the lowest and most economical when compared with that of other power sources. In this scenario, coal power continues to grow until a peak appears at 1741 GW around 2038 and then, from 2035 to 2045, because of the life span of coal power set to 30 years, there is a certain degree of decrease. The total installed capacity of coal power is slightly reduced and eventually is maintained at 1740 GW in the year 2050. Compared with coal power, hydropower and nuclear power have a certain competitiveness for the construction price, and as an important base-load power source, hydropower and nuclear power basically reach the upper limit of exploitable scale in the planning of each planning stage. As an indispensable flexible and peak-regulating power supply, gas power must also be developed in order to support the development and utilization of renewable energy. However, because of the high cost of wind and solar power, the development of renewable energy sources remains insufficient and at an early development stage before 2035; the exploration scale of renewable energy usually reaches the lower limit of the exploitable scale. After 2035, because of the saturation of coal resource development, the increase in coal price, and with the development of new technology, the economy of renewable energy power generation gradually improves. Renewable energy is further developed and utilized, so that renewable energy accounts for around 43% of the total electric power mix by 2050.

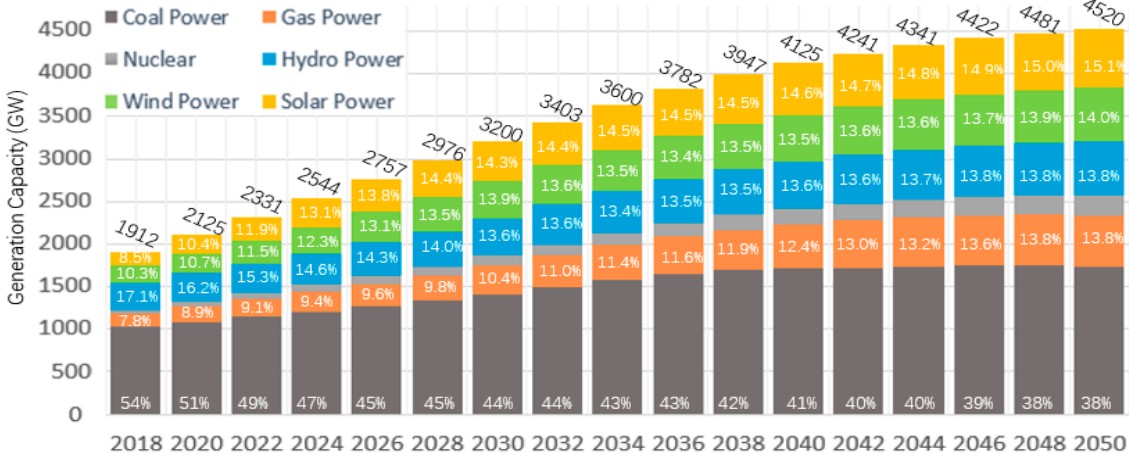

**Figure 7.** China's power structure in S1 from 2018 to 2050.

Figure 8 shows the results of S2, which implements a stringent energy-saving and emissions reduction policy with a coal resource tax, carbon tax, and a tax for other pollutant emissions from 2020. On one hand, according to the planning results of S2, the new installed capacity of coal power grows slowly from 2020 and peaks at 1310 billion kW by 2030. Then, coal use begins to decline gradually from

2030. The remaining coal power capacity is reduced to 960 GW by 2050. On the other hand, renewable energy, especially solar power, benefits from electricity price subsidies, so the cost of power generation shows a rapid decline. The installed scale continues to increase and reaches 420 GW and 830 GW by 2030 and 2050, respectively. In the planning process of S2, hydropower and nuclear power benefit from the lower cost of power generation, which are always prioritized in selection and incorporated into the planning. The stringent energy-saving and emissions reduction policy results in the rapid growth of the cost of coal power, which has a certain inhibitory effect on the large-scale installation of coal power. Finally, under the active and strict policy of energy saving and emissions reduction, China's renewable energy generation capacity accounts for about 54% by 2050.

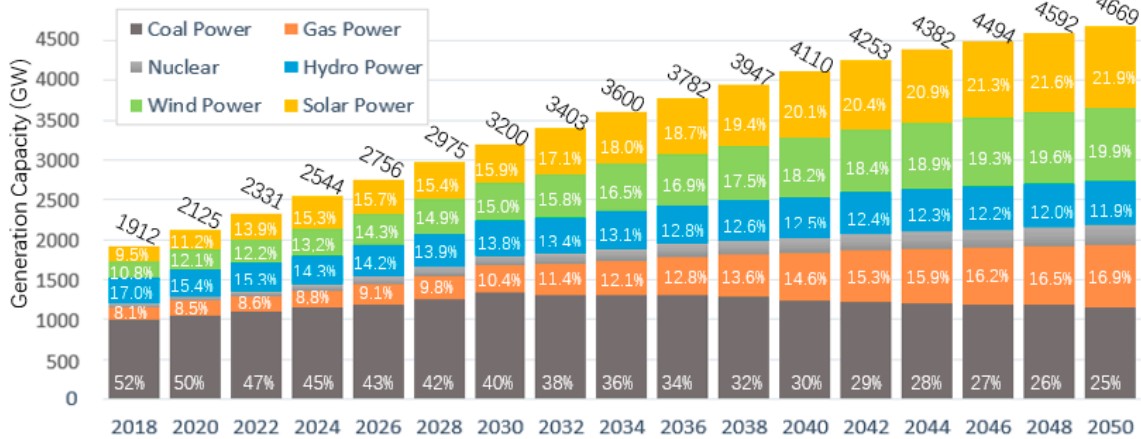

**Figure 8.** China's power structure in S2 from 2018 to 2050.

Figure 9 shows the distribution of coal, gas, wind, and solar power generation in the different regions of China, under the policies of S2 in 2030 and 2050. In terms of thermal power, the layout of coal power installations will mainly be located in north and northwest China. The coal power sources will be more distributed in regions that are rich in coal resources, and their function will be to mainly help to consume and export renewable energy electricity in the region. It should be noted that in S2, the northwest region will be the only region with a momentum of sustained growth in new installed capacity until 2050, and the proportion of 12% will increase to about 18% of the total coal power capacity across all regions. The majority of the new gas power structures will be installed in east and central China by 2030, and in northwest China in the long term. By 2050, the installed gas power capacity in northwest China will account for about 37% of the country's total.

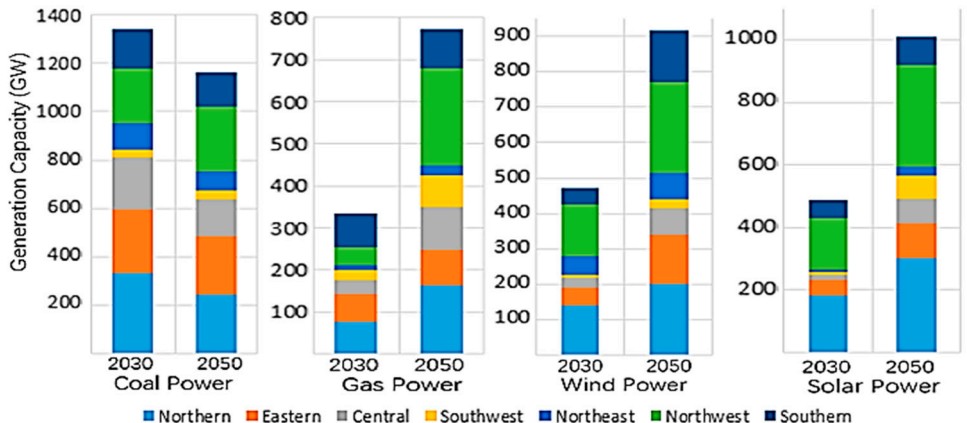

**Figure 9.** Distribution of installations in each region under the policies of S2.

In terms of renewable energy, wind power will still be installed in the northeast, north, and northwest China. The proportion of wind power capacity in the three northern regions will remain above 60% of the country's total for a long time. Although the installed wind power capacity in eastern and southern China will significantly increase after 2030, it is difficult to surpass the three northern regions in terms of total capacity. For solar power, the eastern and central regions still have a large room for growth before 2030. However, in the longer term, the better-resourced northwest China will account for the majority of the growth. This region will account for about 34% of the solar power capacity in China by 2050.

Table 7 above gives the planning results of interregional transmission lines' capacity of S1 and S2. It can be seen that the capacity of the transmission channels will continue to increase during the entire planning period in both scenarios. From the perspective of the growth trend, the capacity of the transmission channels does not slow down with the saturation of power demand. By 2030, 2040, and 2050, China's transmission channel capacity will reach 282, 390, and 461 GW in S1, and 310, 407, and 511 GW in S2, respectively. From the perspective of the cross-regional channel capacity between different regions, northwest to east and central China and southwest to central China are at the forefront in terms of transmission capacity. This further shows that the northwestern and southwestern regions, as resource-rich regions, have obvious sending-end attributes, and east China and central China, as load centers, have obvious receiving-end attributes. In the future, the power flow direction of China will be more evidently from the west to the east.

**Table 7.** Planning results of interregional transmission lines' capacity.

| Transmission Line | S1 | | | S2 | | |
|---|---|---|---|---|---|---|
| | **2030** | **2040** | **2050** | **2030** | **2040** | **2050** |
| North to Central | 5.5 | 10.3026 | 25.72 | 6.05 | 11.32 | 28.81 |
| North to East | 24.44 | 44.528 | 53.00 | 26.88 | 51.83 | 65.69 |
| Northeast to North | 11.81 | 38.05 | 64.29 | 12.99 | 13.41 | 15.00 |
| Northwest to East | 65 | 111.432 | 120.72 | 71.48 | 106.63 | 112.72 |
| Northwest to Central | 100.32 | 117.2985 | 112.95 | 110.32 | 133.12 | 141.72 |
| Northwest to North | 29.1 | 20.223 | 38.31 | 32.00 | 34.92 | 87.30 |
| Central to East | 21.6 | 22.68 | 21.6 | 23.75 | 27.09 | 30.00 |
| Central to South | 24.14 | 25.1056 | 24.14 | 26.55 | 28.23 | 30.14 |
| **Total Cap. (GW)** | 282 | 390 | 461 | 310 | 407 | 511 |

The planning cost comparison of S1 and S2 are shown in Table 8.

**Table 8.** Comparison of planning costs between S1 and S2.

| | Cost ($\times 10^9$RMB) | 2018 | 2020 | 2030 | 2040 | 2050 |
|---|---|---|---|---|---|---|
| | Construction | 2934 | 3191 | 4814 | 9050 | 17,102 |
| S1 | O&M | 8072 | 8970 | 15,234 | 33,997 | 82,569 |
| | Total | 11,006 | 12,161 | 20,048 | 43,047 | 99,671 |
| | Construction | 3298 | 3820 | 6934 | 12,291 | 20,356 |
| S2 | O&M | 11,839 | 13,812 | 29,605 | 58,011 | 96,524 |
| | Total | 15,137 | 17,632 | 36,539 | 70,302 | 116,880 |

According to the planning costs in Table 7, the total social costs of S1 and S2 show an increasing trend year by year. The average annual growth rates for scenarios S1 and S2 are 7.13% and 6.59%, respectively. However, because of the higher initial construction and operation cost of renewable energy, the input cost of S2 in the initial stage is 11.06% higher than that of S1. This cost accumulates to 2050, making the total cost of S2 15.99% higher than that of S1. To be specific, from the construction cost alone, the main power source in S1 is still coal power before 2030, since the planning costs are

relatively stable in construction and operation, and thus the growth rate is stable at about 4.21% per year. After 2030, as the price of coal rises and the price of renewable energy falls, the planning cost rises rapidly with an average annual growth rate of 6.54%. The growth rate in operation costs is similar to that of the construction cost. The growth rate is 5.11% in 2018–2030 and increases to 8.35% in 2030–2050; the total planned cost of S1 by 2050 is 9967.1 billion yuan. The average annual growth rate of the construction and operation costs in S2 steadily decrease year by year. The average annual growth rate falls from 7.93% in 2018 to 5.99% in 2050, and the total planning cost is 11,688 billion yuan by 2050.

Finally, the changes in carbon emissions and pollutant emissions of the two scenarios in the planning period are shown in Table 9.

**Table 9.** Carbon emissions and pollutant emissions in the two scenarios.

|  | Emissions Amount (Billion Tons) | 2018 | 2020 | 2025 | 2030 | 2035 | 2040 | 2045 | 2050 |
|---|---|---|---|---|---|---|---|---|---|
| S1 | $CO_2$ | 3.912 | 4.574 | 5.101 | 5.473 | 5.205 | 5.096 | 4.908 | 4.696 |
|  | $SO_2$ | 0.013 | 0.016 | 0.015 | 0.016 | 0.016 | 0.015 | 0.015 | 0.014 |
|  | $NO_x$ | 0.008 | 0.01 | 0.009 | 0.01 | 0.01 | 0.009 | 0.009 | 0.008 |
| S2 | $CO_2$ | 3.878 | 4.191 | 4.285 | 4.073 | 3.992 | 3.883 | 3.793 | 3.695 |
|  | $SO_2$ | 0.01 | 0.011 | 0.011 | 0.011 | 0.001 | 0.009 | 0.008 | 0.008 |
|  | $NO_x$ | 0.006 | 0.007 | 0.007 | 0.008 | 0.007 | 0.007 | 0.006 | 0.006 |

From Table 8, S1 shows a peak of 5.473 billion tons in the carbon emissions by 2030, which then decreases to 4.696 billion tons by 2050. Carbon emissions in S2 peak at 4.285 billion tons around the year 2025 and then decrease, with an annual growth rate of −0.15% from 2025 to 2050. S2 carbon emissions fall down to 3.695 billion tons by 2050. Other pollutant emissions trends are similar to those of carbon emissions. Overall, the cumulative carbon emissions of S2 are about 31 billion tons less than those of S1 over the 35-year planning period from 2018 to 2050.

## 5. Conclusions

Low-carbon and clean transition now represent the main development direction of China's electric power sector. In order to achieve the sustainable transition target of power industry development, the Chinese government has formulated a series of policies and plans regarding energy savings and emissions reduction to control and guide the behavior of participants in China's electric power sector. These indicative, macro-level planning and development plans require the cooperation of professional power system analysis tools that can decompose the macro-level requirements into a specific power system state and corresponding technical economic indexes and further specific implementations. However, previous power system planning and analysis tools are usually static analysis models with all parameters fixed and do not effectively reflect or evaluate the influence of power technology diffusion and incentive policy changes on the process of power system transition. Therefore, from the perspective of the sustainable transition theory and low-carbon goals of China's power development, an MDP-based dynamic IG-TEP model, in which the state transfer matrix is formed by levelized power generation cost that considers power technology learning and the influence of a low-carbon incentive policy scenario, is introduced in this paper.

The MDP-based dynamic IG-TEP model takes power demand, the technical characteristics of power supply, and transmission networks as the endogenous variables that change with the stage. We used the current regional power system structure, the regional resource endowments, and information of existing transmission channels as the initial input data and considered constraints such as power reliability, transmission capacity, generation and transmission expansion limitations, renewable energy penetration, carbon emission limits, and other pollutant emissions limits. The minimization of the total planning cost in the planning period was used as the objective function to optimize the installed

power structure, transmission channel capacity, planning costs, and emissions of carbon dioxide and other pollutants.

By the generation and transmission integrated expansion model, this paper analyzed the transition pathway of China's power sector from 2018 to 2050 under different policy intensities. The analysis showed that if the intensity of energy-saving and emissions reduction policies in 2017 is maintained, the inhibition on the large-scale development of coal power will not be obvious, and coal power will peak at 1.33 billion kW around the year 2036. Although there will be a retirement of coal power from 2035 to 2045, the development of renewable energy power will still remain relatively slow, and the renewable energy capacity will account for 43% of the total electric power capacity. In the case of the implementation of strict energy-saving and emissions reduction policies, the development of coal power will be significantly inhibited from 2020, will reach a peak of 1.16 billion kW in 2026, and will then gradually decline to 960 million kW by 2050. At the same time, the scale of solar power will continue to increase due to policy incentives, reaching 420 million kW in 2030 and 830 million kW in 2050. In the scenario of strict energy-saving and emissions reduction policies, China's renewable energy capacity will increase to 54% by 2050. However, limited by resource endowments, hydropower and nuclear power will basically reach the upper limit of the exploitable scale in the later planning stage in both scenarios, and changes in their planned capacities after 2030 are not obvious.

**Author Contributions:** Methodology, H.H.; Writing—Original Draft Preparation, H.H., and L.L.; Data Collection, Z.T., and L.L.; Writing—Review & Editing, all the authors.

**Funding:** This research was funded by National Natural Science Foundation of China under Grant No.71774039.

**Conflicts of Interest:** The authors declare no conflict of interest.

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
