# Peer review of "Research on China’s Power Sustainable Transition Under Progressively Levelized Power Generation Cost Based on a Dynamic Integrated Generation–Transmission Planning Model"

_sustainability, doi:10.3390/su11082288_

Round 1

Reviewer 1 Report

Low carbon transformation will become the main development direction of China’s electric power sector in the future. In this paper, a dynamic generation and transmission integrated planning model based on the Markov decision process was established as a macro-level power system analysis tool to analyze the effect and influence of energy savings and emissions reduction policies. This model takes power demand, the technical characteristics of power supply, and transmission networks as the endogenous variables that change with the stage. We use the current regional power mix, the regional resource endowments, and information of existing transmission channels as the initial input data. Some suggestions for improving this study are as follows:

In section 3.2 “The emission coefficient and annual emission limitations of fired power sources are shown in Table 2. The data and parameters in Table 2 were obtained from References [30,31] or calculated according to the material content.” The authors should provide more evidences and existing data to illustrate the rationality of initial simulation data.

The drawings in this paper, such as Figure 7 and Figure 4, are blurred and not refined enough. In the figure 4, “eight transmission channels were constructed to connect the six grid regions”. Are the eight transmissions same scale? Are these only unidirectional transmission? Please explain

The authors proposed a dynamic power-grid integrated planning model to describe the power demand sequence parameters, and the characteristic parameters of different generation equipment and transmission grids. Why are these parameters selected? The authors should provide more evidences and existing literatures to illustrate the rationality of these selections.

In section “the low carbon policy portfolio can play a key role in guiding the development of a power system,” How does policy affect the low-carbon development of power system as a tool? Nevertheless, the policy regard as a constraint condition, such as “this paper analyzed the development of China’s power sector from 2018 to 2050 under different policy intensities.” The authors should provide more evidences to analyse the policy.

Author Response

Dear Reviewer,

First of all, thank you for your patience and tolerance of the problems and mistakes in the article. Second, according to your review opinions, the problems in this paper can be summarized into three aspects:

1.     In this paper, the purpose and significance of establishing the model and the research method selection are not clearly explained;

2.     The results analysis and conclusions of this paper are not very clear;

3.     Some content is complicated and the details are unclear.

So, according to the above opinions, the article is modified as follows:

1.     For the ambiguous about the topic and the characteristics of the model, new contents have been added in the part of introduction, which the main concern in this paper is‘ the transition pathway of China’s power sector under the impact of the levelized power generation cost change’. The method selected is the integrated power generation-transmission planning model based on Markov decision process. And the state transition function of Markov decision process is composed of the levelized generation cost model considering learning curve and incentive policy scenarios.

2.     According to the theme direction, the content of the result analysis and conclusion part has been rewritten. The results analysis section revolves around the two set scenarios (where, S1 is remain current level reduction policy and S2 is Strict reduction policy). And the results could conclude as in given information, the coal power will maintain its dominant position in the power structure during the planning period in S1; and renewable power will develop rapidly and gradually replace coal power as the most important power source in the power structure from 2030 in S2.

3.     According to the above content, the title of the article has been adjusted accordingly. The adjusted title is: The Research of China’s Power Sustainable Transition Under the Progress of Levelized Power Generation Cost Based on Dynamic Integrated Generation-Transmission Planning Model.

4.     The layout of the article has been adjusted and the main content has been simplified, the possible ambiguous content is further explained as well.

Finally, thanks for your valuable comments, And I hope the above modifications can answer your questions about the paper and could help for better understanding.

Reviewer 2 Report

The article presents a valuable analysis of China's attempt to transfer from coal to renewable energy in the decades up to 2050. Data analysis and scenarios are used to envisage the attempt to reduce the use of coal and increase the use of renewable energy.

However, although the data and methodology are presented clearly in the body of the paper, the conclusions could be made clearer. For instance, in the last paragraph of the paper what exactly does "there will be a retirement of coal power from 2035 to 2045" mean? This suggests that coal power will cease to be used, whereas it seems as if the authors mean to say that there will be a reduction in the use of coal power. The word 'retirement' is confusing and seems to contradict the statement in the abstract (and introduction) that "China’s power development will maintain a coal-oriented power development path as the space for renewable energy is limited."

So my main criticism is that it is difficult to understand the overall conclusions of the analysis, especially in the final paragraph of the paper and how it links to the abstract and introduction. The research findings need to be drawn out much more clearly in the conclusion, and the abstract and introduction should match the conclusion. I think the main thing that readers want to know is whether the use of coal is projected to significantly reduce or not. Are renewables going to replace coal to a great extent or not? The conclusion is not totally clear about this most crucial point. So I would suggest improving the text of the existing final paragraph as well as adding another paragraph to clarify the situation.

Author Response

(The authors gave the same response as above.)

Reviewer 3 Report

The subject matter of this paper is quite interesting and relevant for sustainability transitions. The idea of integrating the power extension and generation extension by taking a dynamic planning perspective is logical. However, I see two main points for improvements: first, regarding the method, it would be beneficial to argue why you have used it, especially I believe you can compare it to other methods currently being used in the literature, especially system dynamics modeling. For instance:

Pereira, A. J., & Saraiva, J. T. (2011). Generation expansion planning (GEP)–A long-term approach using system dynamics and genetic algorithms (GAs). Energy36(8), 5180-5199.

For GEP, and: 

Dehdarian, A. (2018). Scenario-based system dynamics modeling for the cost recovery of new energy technology deployment: The case of smart metering roll-out. Journal of Cleaner Production178, 791-803.

for an idea of how to combine different scenarios.

Second, the model validation and sensitivity analysis is missing and suggest to add a section before conclusion to discuss the results based on the validity of the assumptions and changes in initial conditions of the scenarios.

Some minor changes in the presentation would be useful, for instance the quality of figure 4 can be improved and there can be an index for all the acronyms used in the text and in the equations to summarize them, instead of explaining them in the text. 

Author Response

(The authors gave the same response as above.)

Round 2

Reviewer 1 Report

This article has already revised in accordance with my suggestions. Now, I suggest it could be publication.